# Carbon-Infiltrated Carbon Nanotube Topography Reduces the Growth of *Staphylococcus aureus* Biofilms

**DOI:** 10.3390/nano15070510

**Published:** 2025-03-28

**Authors:** Lucy C. Bowden, Sidney T. Sithole, Anton E. Bowden, Brian D. Jensen, Bradford K. Berges

**Affiliations:** 1Department of Microbiology and Molecular Biology, Brigham Young University, Provo, UT 84602, USA; lucycg6@student.byu.edu (L.C.B.); sydyten@byu.edu (S.T.S.); 2Department of Mechanical Engineering, Brigham Young University, Provo, UT 84602, USA; abowden@byu.edu (A.E.B.); bdjensen@byu.edu (B.D.J.)

**Keywords:** biofilm, carbon nanotubes, MRSA, *Staphylococcus aureus*, nanostructured surface

## Abstract

Orthopedic implant-associated infections are a growing problem. These infections are often associated with bacterial biofilms, such as those formed by *Staphylococcus aureus*. Nanotextured surfaces can reduce or prevent the development of bacterial biofilms and could help reduce infection rates and severity. Previous work has shown that a carbon-infiltrated carbon nanotube (CICNT) surface reduces the growth of *S. aureus* biofilms. This work expands on previous experiments, showing that the topography of the CICNT, rather than its surface chemistry, is responsible for the reduction in biofilm growth. Additionally, the CICNT surface does not reduce biofilm growth by killing the bacteria or by preventing their attachment. Rather it likely slows cell growth, resulting in fewer cells and reduced biofilm formation.

## 1. Introduction

Orthopedic implants are very common medical interventions that greatly improve patient quality of life. Over a million hip and knee replacements are performed each year in the United States alone [1], with a projected increase to nearly two million per year in Medicare patients alone by 2030 [2]. However, implant-associated infection sometimes follows these surgeries, accounting for about 23% of all-cause revision surgeries [3]. These infections are extremely difficult to treat because they are often associated with bacterial biofilms, which are resistant to clearance efforts by both the human immune system and antibiotic treatment [4,5]. *Staphylococcus aureus* is a biofilm-forming Gram-positive bacterium that is the most frequent causative agent of implant-associated infections [6].

Alternative methods of preventing implant-associated infection could help alleviate issues with both treating biofilm-associated infections and preventing the spread of antibiotic resistance. One potential alternative is device surface coatings that help resist bacterial colonization. The topography of nanotextured surfaces such as those found on insect wings can prevent the formation of bacterial biofilms [7]. This effect can also be found on synthetic surfaces [8].

Carbon-infiltrated carbon nanotubes (CICNTs) are a surface coating that has previously been shown to have some antimicrobial properties, reducing the growth of *S. aureus* biofilms [9,10]. Carbon nanotubes are cylindrical molecules of bonded carbon atoms. A post-processing carbon infiltration step adds bulk carbon to the nanotubes, greatly increasing their diameter. In the previous study by Morco et al., bacterial biofilms on CICNT surfaces were measured by SEM imaging and manual cell counting. The goal of this study was to investigate how the CICNT surface reduces biofilm formation in a more physiologically relevant medium using colony forming unit bacterial quantification. Using multiple controls, we determined that the surface inhibits bacterial adhesion primarily through its topography rather than its chemical properties. Additionally, our findings suggest that the surface reduces biofilm formation by limiting bacterial growth rather than by killing the cells or preventing initial attachment. This study introduces a novel surface with anti-biofilm properties that may uniquely inhibit *S. aureus* biofilm formation.

## 2. Materials and Methods

### 2.1. CICNT and Control Sample Preparation

Surfaces coated in carbon-infiltrated carbon nanotubes were prepared as described previously [10]. Briefly, medical grade Ti6Al4V alloy (Ti) was cut from sheet stock into 9 mm squares (herein referred to as samples). Samples were sonicated in isopropyl alcohol for 15 min, rinsed in deionized water, and dried. For the preparation of CICNT samples 200 nm of Al_2_O_3_ was deposited on the surface of each sample as a barrier layer using electron-beam deposition. This was followed by the deposition of a 6 nm thin film of iron as a catalyst layer using a thermal evaporator.

The prepared samples were then placed into a tube furnace for CICNT growth, which was performed as described previously [10], with the infiltration step resulting in nanotubes with a final diameter of approximately 150 nm, which was confirmed using scanning electron microscopy (SEM). Carbon control samples were prepared in a similar manner, with the presence of the Al_2_O_3_ layer, but without the iron layer. They were then placed into the furnace and a carbon deposition step was completed, resulting in a surface coated with carbon but without distinct morphology. To prepare Al_2_O_3_-coated samples, bare Ti and prepared CICNT samples were coated with 10 nm of Al_2_O_3_ using atomic layer deposition (ALD150LX, Lesker, Jefferson Hills, PA, USA). This addition did not alter the surface topography but did cover the entire surface in a thin layer of Al_2_O_3_.

### 2.2. Bacterial Strains

JE2 (BEI Resources NR-46543, Bethesda, MD, USA) is derived from USA300 LAC, a well-characterized methicillin-resistant *S. aureus* strain isolated from the Los Angeles County jail in 2002 [11]. JE2 differs from the parent LAC strain by the removal of two cryptic plasmids [12]. SH1000 (BEI Resources NR-55396) is a methicillin-sensitive human isolate of *S. aureus* with the *rbs*U gene restored. SA29213 (ATCC, 29213, Manassa, VA, USA) is a methicillin-sensitive clinical wound isolate. HA2 is a methicillin-resistant clinical isolate donated from a local hospital pathology lab.

### 2.3. Bacterial Culture

Cultures of *S. aureus* were grown in shaking culture for 16–18 h in tryptic soy broth (TSB). All cultures were then diluted to an optical density of 0.02 in RPMI 1640 media, supplemented with sodium bicarbonate and HEPES (4-(2-hydroxyethyl)-1-piperazineethanesulfonic acid), as well as 2% heat-inactivated fetal bovine serum (FBS). A total of 25 μL of inoculated RPMI was pipetted as a droplet onto the top of each sample. This method of droplet growth was performed as previously reported [10]. The droplets of bacterial culture were then incubated on the surface of each sample at 37 °C for the given amount of time. One overnight shaking culture was used to inoculate 2–3 separate samples of each surface type. Each of these samples was then analyzed separately. This entire format was repeated 3–4 times to ensure reproducibility, resulting in at least 6 replicates for each experiment.

### 2.4. Biofilm Quantification

Multiple methods of biofilm quantification were considered, including crystal violet and CFU quantification. Crystal violet is often considered the gold standard method, but the dye becomes trapped in the porous nanotubes on the CICNT surface, producing substantial background staining that confounds the results. Therefore, we used serial dilutions and colony forming unit (CFU) quantification. CFU analysis was performed as reported previously [10]. Briefly, after 36 h (unless otherwise indicated) of incubation in 24-well plates, the samples with attached biofilms were washed once in sterile 1× phosphate-buffered saline (PBS) to remove unadhered cells. Samples were then removed to a sterile well for biofilm removal. An amount of 500 μL of PBS was added to the sample surface and pipetted vigorously to dislodge the biofilm, followed by vortexing for 1 min. This process effectively detached the adherent cells from all surfaces. Then, 10 μL from each well were then removed and serially diluted in PBS before being spread on LB agar plates [13,14]. Results represent the average of three independent experiments, each consisting of biofilms grown on two samples per surface type. Each disturbed biofilm was diluted and plated onto a single agar plate, resulting in a total of six replicates per condition to ensure reproducibility. The plates were then incubated at 37 °C for 24 h, after which colonies were counted.

### 2.5. Propidium Iodide Staining and Flow Cytometry

After biofilms had been grown on samples for 36 h, the biofilms were disrupted by pipetting using 500 μL of PBS. This was performed without an initial washing step so that the final solution contained both unattached bacteria and bacteria from the biofilm. Next, 4 μL of 2.5 mg mL^−1^ propidium iodide (PI, Sigma-Aldrich, Saint Louis, MO, USA) was added to the bacterial suspension, mixed, and then incubated in the dark for 15 min at room temperature. Immediately following the 15 min incubation, the cells were analyzed via flow cytometry. Positive controls for dead cells were prepared by incubating bacteria in 70% isopropyl alcohol for 30 min at room temperature. An unstained control was used as a negative control to guide the gating process. The fluorescence of the PI stain was analyzed on a CytoFLEX flow cytometer (Beckman Coulter, Brea, CA, USA) and data were analyzed using FlowJo software (version 10.6.2). The gating strategy was informed by the positive and negative controls. Results represent the average of two separate experiments, each consisting of three individual samples of each surface type, for a total of six replicates.

### 2.6. Scanning Electron Microscopy

Scanning electron microscopy (SEM) images were taken on a ThermoScientific Verios G4 UC SEM (Waltham, MA, USA). Samples of each surface type were imaged prior to biofilm formation using either a flat stub at an accelerating voltage of 5 kV and ion beam current of 50 pA or at an 80° tilt with an accelerating voltage of 10 kV and an ion beam current of 0.1 nA.

### 2.7. Statistical Analysis

Comparisons of CFU data were analyzed using a single-factor ANOVA test, followed by a Tukey–Kramer post hoc analysis. Flow cytometry data were analyzed using a single-factor ANOVA test. CFU data from the time course experiment were analyzed using Student’s *t*-test. Statistically significant differences were attributed to variables with *p* ≤ 0.05.

## 3. Results and Discussion

### 3.1. The CICNT Surface Reduces S. aureus Biofilm Growth in RPMI Media

RPMI inoculated with *S. aureus* was incubated for 36 h on either bare titanium (Ti), carbon, or CICNT surfaces. In our previous work, we performed similar experiments but with TSB growth media used during biofilm growth. In the experiments described below, we used RPMI supplemented with fetal bovine serum to study biofilm development in an environment that more closely mimics that of human patients while still promoting adhesion and biofilm formation of *S. aureus* [15]. Ti was used as a control because titanium is a very common implant material. The carbon surface was used as a secondary control because it differs from the CICNT surface in texture, but not in the presence of carbon. SEM imaging was used to show the topography of each surface. Images were taken both top-down and at an 80° tilt. These images showed that the topography of the Ti surface and the carbon surface were virtually indistinguishable, while the topography of the CICNT surface was distinctly different and more textured (Figure 1).

Four different isolates of *S. aureus* were tested on all three surface types, and biofilms were quantified using CFU analysis after serial dilutions. An ANOVA on each isolate showed that there was significant variation among the surface types (JE2: *p* = 1 × 10^−7^, HA2: *p* = 0.001, SH1000: *p* = 0.0001, SA29213: *p* = 0.04.) Post hoc Tukey–Kramer tests showed that there was no significant difference between the Ti and carbon control groups for any isolate, suggesting that the presence of carbon is not responsible for any biofilm reduction (Figure 2). For JE2, HA2, and SH1000, there was a significant reduction in the number of bacteria on the CICNT surface compared to both the Ti and carbon surfaces (Figure 2). These fold changes in the number of bacteria on Ti and CICNT ranged from a 3.7-fold reduction in the SH1000 strain to a 3.4-fold reduction for the JE2 strain to a 2.1-fold reduction for the HA2 strain. For SA29213, the reduction in the number of bacteria on the CICNT surface compared to the Ti surface was not significant (1.4-fold reduction), but there was a significant reduction in the number of bacteria on the CICNT surface compared to the carbon surface (1.6-fold reduction). This discrepancy is likely due to the small reduction in bacteria overall and a larger spread of the data on Ti surfaces for this isolate.

In our previous study using TSB as our bacterial growth media, we had found that there was a significant 2.5-fold reduction in the number of adherent bacteria from the JE2 strain on CICNT compared to Ti, which was confirmed using both CFU counts and SEM images of the biofilm. Additionally, we found that there was a small but significant 1.5-fold reduction in the number of adherent bacteria on the carbon control compared to Ti [10]. However, in this study where we used the more physiological RPMI as the growth media, there was no significant difference in the number of adherent bacteria recovered from the Ti and carbon surfaces. In both studies, there were fewer bacteria recovered from the CICNT surface than either the Ti or carbon surfaces, suggesting that the surface texture may affect biofilm formation. Interestingly, we had also previously found that the strain HA2 uniquely grew significantly stronger biofilms on CICNT vs. titanium [10]. Our current results show the opposite effect (2.1-fold reduction on CICNT). Additionally, in that study we found that there was a significant reduction in the number of bacteria on Ti vs. CICNT for the SA29213 strain, but in this study the difference was not significant. These differences may be due to the different compositions of the growth media. Others have previously reported that biofilm composition and strength differ for a single isolate of bacteria based on nutritional options in the growth media [16,17,18]. Since many experiments into the antimicrobial properties of various surfaces use bacterial growth media such as TSB, it is of note that antimicrobial surfaces may behave differently in different media, including those that provide more physiological conditions.

### 3.2. The CICNT Topography, Rather than the Presence of Carbon, Reduces Biofilm Growth

The antimicrobial properties of the CICNT surface might arise due to the surface chemistry or due to the physical topography of the surface. In the previous experiment, we created an amorphous carbon control, which was covered in a surface that had a similar surface chemistry to the CICNT surface, but a distinctly different topography that was more similar to bare Ti. As was shown in Figure 2, no significant difference was found in the number of adherent bacteria recovered from Ti or carbon, and significantly less (3.4-fold) bacteria were recovered from the CICNT surface using the JE2 strain. This indicated that the presence of carbon by itself was not antimicrobial, and suggested that the specific shape of the CICNT surface was instead responsible. To confirm this, we utilized a second set of samples where the surfaces of both Ti and CICNT were coated in a 10 nm layer of Al_2_O_3_, resulting in two samples with the same material on the surface, but two different topographies (Figure 3A). Al_2_O_3_ is a biologically inert ceramic material commonly used for implants [19]. The CICNT surface has nanotubes with an average diameter of 150 nm, and the Al_2_O_3_-CICNT surface nanotubes were not significantly different in diameter. There was a significant reduction in bacteria recovered from the Al_2_O_3_-CICNT compared to the Al_2_O_3_-Ti. Although fewer bacteria overall were recovered from Al_2_O_3_-coated surfaces than non-coated surfaces, the ratio between bacteria recovered from Al_2_O_3_-Ti and Al_2_O_3_-CICNT (3.8-fold, Figure 3B) was very similar to the ratio recovered from Ti and CICNT. This suggests that the surface topography itself, rather than the chemistry of the material present at the bacteria–surface interface, may be reducing biofilm growth.

A similar experiment was performed on cicada wings by Ivanova et al. [20]. Cicada wings possess a topography that results in the death of bacterial cells. To confirm that the insect wing topography was responsible, rather than its surface chemistry, the authors coated the wing surface with 10 nm of gold, altering the surface chemistry while retaining its topography. Like in our experiment, the antibacterial effect persisted, suggesting that the surface topography plays an important role in how many bacteria can attach.

### 3.3. The Reduction in Biofilm Growth on CICNT Is Not Due to Bacterial Cell Death

A reduction in adherent bacteria on a surface can be caused by cell death, a lack of attachment, or by some other mechanism. Many surfaces, such as insect wings, prevent biofilm growth by killing bacteria [7,8]. To determine whether the CICNT surface was killing bacteria, biofilms were grown for 36 h on Ti, carbon, and CICNT surfaces. The cells (both adherent and non-adherent) were then removed, and propidium iodide (PI) was added to stain for the presence of dead cells. PI is a membrane-impermeable stain that binds to nucleic acids. Quantifying PI-positive cells can then be used as a measure of cell death. Flow cytometry was used to quantify the percentage of PI-positive cells. No significant difference was found in the percentage of PI-positive cells between any of the three surfaces (Figure 4, *p* = 0.31). This finding suggests that cell death is not a major contributing factor to the reduction in bacteria seen on the CICNT surface, and therefore the CICNT surface is likely not bactericidal.

### 3.4. Mid to Late Biofilm Growth Is Slower on CICNT than on Ti Surfaces

Some surfaces reduce biofilm growth by preventing bacterial attachment, rather than by directly killing the cells [21]. Another mechanism of biofilm reduction could be reduced cell growth. To determine whether these were at play on the CICNT surface, biofilm growth was measured by CFU quantification at several time points (1, 12, 24, and 36 h). After one hour, there was no significant difference in the number of adherent bacteria on Ti or CICNT (*p* = 0.12, Figure 5). This finding was similar at 12 hours, where there was also no significant difference. These results suggest that a difference in initial attachment is not the mechanism behind the reduction in biofilm recovery when only analyzed at a single, later time point. However, at 24 and 36 h, there was a significant and increasing difference in the number of adherent cells (2-fold reduction on CICNT at 24 h, 3.6-fold at 36 h, Figure 5). This suggests that one possible mechanism for the reduction in adherent bacteria on the CICNT surface at later time points may be due to reduced cell growth. To further test this hypothesis, we also quantified the unattached cells at multiple time points because a decrease in attached cells could also be due to biofilm dispersal. This experiment showed that there was not a marked rise in non-adherent cells at the 24 h or 36 h time points (Figure 5). This suggested that the reduction in biofilm growth on the CICNT surface was not due to a biofilm dispersion event. It also confirmed that the reduction in biofilm growth is unlikely to be due to inhibition of attachment. The cause of the general reduction in cells on both surfaces at 36 h may be due to a number of factors, including nutrient restriction in the droplet growth method.

Commonly, materials exhibiting structural antibiofilm properties are considered in one of two ways. They may be bactericidal (killing the bacteria on contact), or antibiofouling, which is generally defined as the prevention of bacterial attachment [22]. For example, many insect wings exhibit bactericidal properties [7,20], while shark skin is considered to be antibiofouling [21]. However, the CICNT surface does not fit neatly into these two categories and may act in a more novel manner. Flow cytometry data indicate that the bacteria are not dying more rapidly on a CICNT surface than on a Ti surface, and CFU data indicate that at 1 h and 12 h there is no difference in the number of bacteria on the CICNT surface and a Ti surface, suggesting no difference in initial attachment rates. Rather than being bactericidal or antibiofouling, the CICNT surface may fall into a less-explored third category, where the surface topography restricts bacterial growth without killing the bacteria even though the bacteria can effectively attach. Surfaces of this type could be referred to as bacteriostatic if they restrict bacterial growth or replication but do not kill the bacteria.

Few examples of bacteriostatic surfaces exist in the literature, but one example of a surface that reduces bacterial growth in this manner was described by Sorzabal-Bellido et al. In their work, the authors used a surface coated in vertically aligned silicon nanowires [23]. When they grew *S. aureus* on the surface, they found that the surface topography restricted the bacterial growth. This may be due to the way that *S. aureus* divides. Cell division in *S. aureus* occurs only along a perpendicular plane to the previous plane of division. Therefore, a textured surface can provide barriers to future division planes resulting in an overall reduction in bacterial growth.

One interesting facet of the data presented in Figure 5 is that there is a decrease in planktonic cells as well as adherent cells on the CICNT samples at 36 h. One possible explanation for this could be that certain areas of the CICNT surface are more amenable to bacterial growth than others, either because of intrinsic antibiofilm factors or because of a physical restriction on bacterial division. Bacteria may be able to settle and grow into mature biofilms on some areas of the CICNT surface, while on other areas of the surface they can settle but their growth is restricted, resulting in a lack of mature biofilms. The mature biofilms would proliferate and also release bacteria back into the surrounding media during dispersion events. However, the bacteria that were unable to form biofilms might neither proliferate nor release large quantities of bacteria. This discrepancy could explain the difference in the number of cells both in the biofilm and in the surrounding media compared to a titanium surface where the bacteria can settle and form biofilms anywhere on the available surface. Although further testing would need to be conducted to confirm or refute this hypothesis, some limited evidence shows that in SEM images of mature biofilms on the CICNT surface the biofilm seems to form in more isolated tower-like structures compared to completely coating a titanium surface [10].

In addition to mechanical barriers to replication, it is possible that textured surfaces such as CICNT could induce gene expression changes that reduce bacterial growth rates through mechanical stresses exerted by the surface. Bacteria are sensitive to mechanical forces and signals, including fluid shear, surface stiffness, surface porosity, and forces induced by bacterial attachment [24,25]. Attachment to surfaces with varying mechanical properties has been shown to cause changes in virulence [26] and biofilm development in *Pseudomonas aeruginosa* [27]. It can also influence rates of cell growth and division. For example, surface stiffness was found to influence the rate of colony spreading for *Serratia marcescens* biofilms [28]. Stress resulting from surfaces physically crowding *Bacillus subtilis* colonies was also shown to reduce growth and induce spore formation [29].

Although much research in this field remains to be carried out, it is plausible that surface texture could induce cellular changes that reduce *S. aureus* biofilm growth. There are some limitations in the current work. Notably, CFU quantification only accounts for metabolically active cells, potentially overlooking dormant or non-culturable populations. Additionally, as these experiments were performed in vitro, the CICNT surface may behave differently in vivo. Future studies will address these issues and further investigate the mechanisms of the CICNT antibiofilm effect. Metabolic activity studies could be used to confirm a reduction in cell growth. Transcriptomic analyses of cells grown on CICNT surfaces and bare Ti surfaces could reveal transcription-level changes to cell growth and activity. Flow cytometry could be used to confirm differences in cell division rates, while microscopy studies could also help to better understand whether specific regions of the CICNT surface preferentially support biofilm growth. This is an open area of research that merits further investigation.

## Figures and Tables

**Figure 1 nanomaterials-15-00510-f001:**
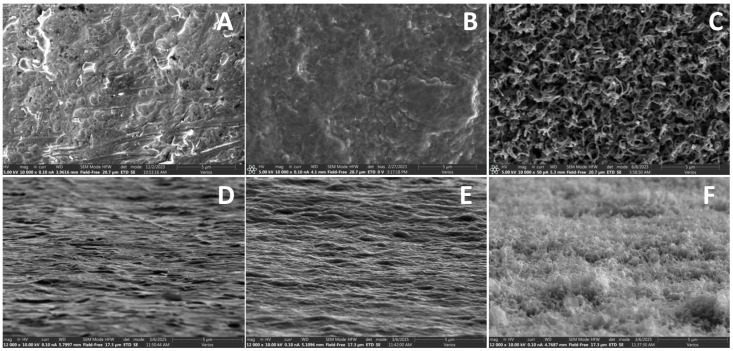
SEM images of top-down SEM images of (**A**) bare Ti, (**B**) carbon control, and (**C**) CICNT at 10,000×. Side view (80° tilt) SEM images of (**D**) bare Ti, (**E**) carbon control, and (**F**) CICNT at 12,000×.

**Figure 2 nanomaterials-15-00510-f002:**
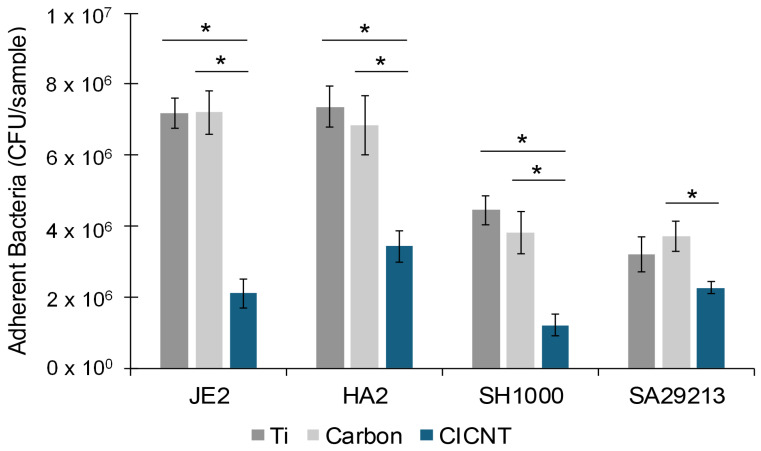
CFU/sample +/− standard error for adherent bacteria from four *S. aureus* isolates grown on either bare Ti, unstructured carbon, or CICNT surfaces. One overnight shaking culture was used to inoculate 2–3 separate and independent samples of each surface type. Each of these samples was then analyzed separately. This entire format was repeated 3–4 times to ensure reproducibility, resulting in at least 6 replicates for each isolate. Biofilms were grown for 36 h in RPMI. The difference in bacteria between Ti and carbon surfaces was not significant for any isolate. * *p* < 0.01.

**Figure 3 nanomaterials-15-00510-f003:**
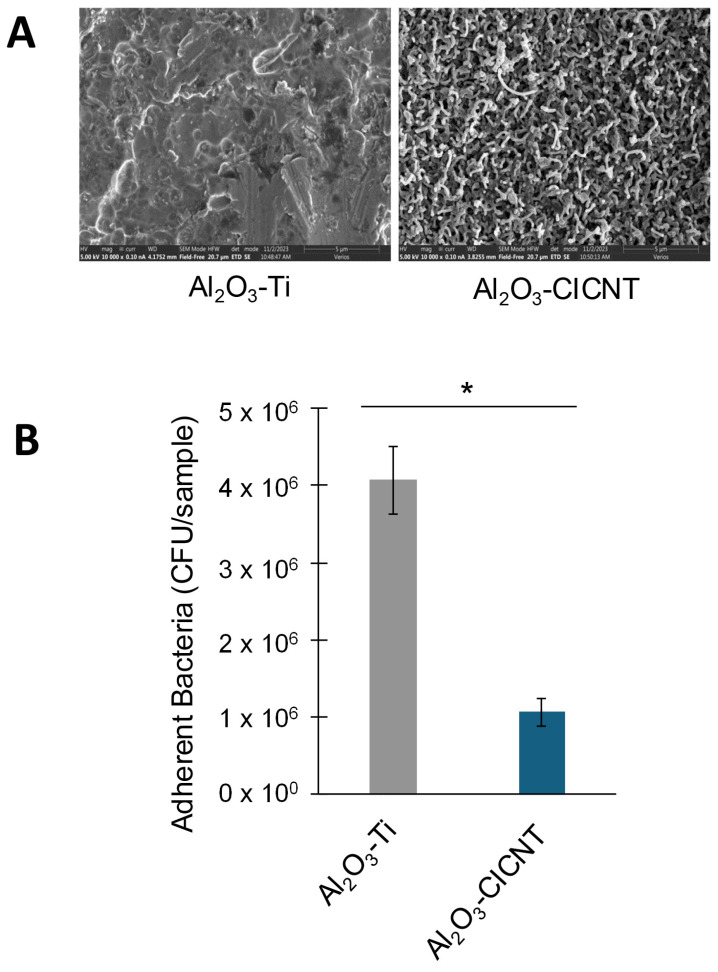
(**A**) SEM images of Al_2_O_3_-coated Ti and Al_2_O_3_-coated CICNT surfaces before biofilm growth. (**B**) CFU/sample +/− standard error for adherent bacteria from *S. aureus* (JE2) grown on Al_2_O_3_-coated Ti or Al_2_O_3_-coated CICNT surfaces. One overnight shaking culture was used to inoculate 2–3 separate and independent samples of each surface type. Each of these samples was then analyzed separately. This entire format was repeated 3 times to ensure reproducibility, resulting in 8–9 replicates for each surface type. Biofilms were grown for 36 h in RPMI. * *p* < 0.01.

**Figure 4 nanomaterials-15-00510-f004:**
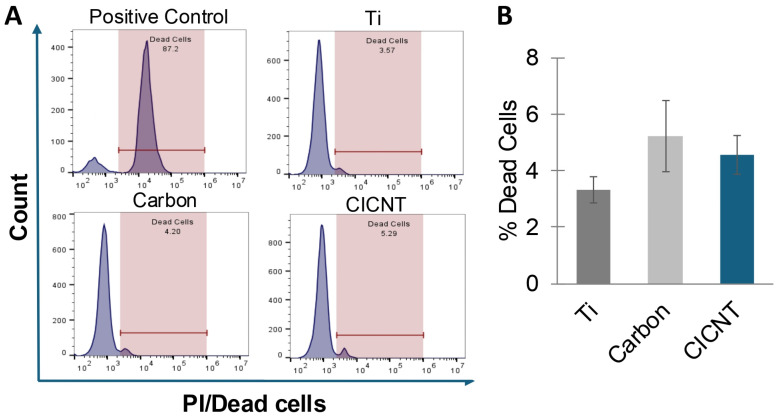
(**A**) Representative flow cytometry histograms for PI-positive cells from bacteria grown on Ti, unstructured carbon, or CICNT. The positive control consists of cells that were grown in a biofilm and then killed using 70% isopropyl alcohol. (**B**) Percent of dead cells after 36 h on various surfaces as measured by flow cytometry using PI to show membrane-permeable cells. Differences between groups are not significant (*p* = 0.31). Measurements are from combined adherent and non-adherent cells after 36 h of growth in RPMI (JE2 strain). One overnight shaking culture was used to inoculate 3 separate and independent samples of each surface type. Each of these samples was then analyzed separately. This entire format was repeated 2 times to ensure reproducibility, resulting in 6 replicates for each surface type.

**Figure 5 nanomaterials-15-00510-f005:**
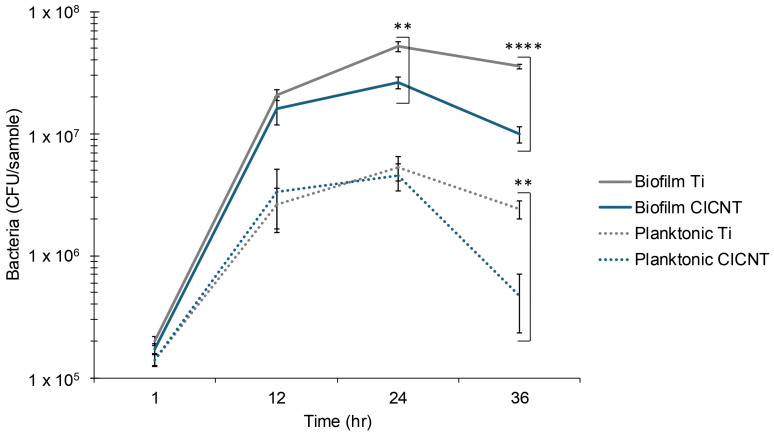
CFU/sample +/− standard error for adherent *S. aureus* (JE2) grown on bare Ti or CICNT surfaces. Biofilms were grown for the given amount of time in RPMI. One overnight shaking culture was used to inoculate two individual samples of either CICNT or Ti. Each sample was analyzed separately. This process was repeated on new samples three times to ensure reproducibility, for a total of 6 samples of both CICNT and Ti at each time point. **** *p* < 1 × 10^−9^, ** *p* < 0.001.

## Data Availability

The data presented in this study are available on request from the corresponding author.

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
