# Peer review of "Carbon-Infiltrated Carbon Nanotube Topography Reduces the Growth of Staphylococcus aureus Biofilms"

_nanomaterials, 2025, doi:10.3390/nano15070510_

Round 1
Reviewer 1 Report
Comments and Suggestions for Authors
In this study, authors studied the effect of Carbon-Infiltrated Carbon Nanotube coating of Ti surfaces on biofilm formation and proposed an explanation about its mechanism. The study showed some interesting results, but it does not significantly improve the findings previously published by the authors (Bowden LC et al. (2023) Carbon-infiltrated carbon nanotubes inhibit the development of Staphylococcus aureus biofilms. Sci 338 Rep 13:19398 doi: 10.1038/s41598-023-46748-y). In the present manuscript, the difference was that authors used a different growth media and tried to identify if the bacteria released from biofilms were still alive or dead by means of a flow cytometry experiment.
However, I think that some of the conclusions are still hypotheses and could be better clarified.
As the authors suggest that the topography of the surface was responsible for the reduction of biofilm formation, it is of outmost importance to present a clear image of the coated surface compared to the uncoated one. Comparison with amorphous carbon coating is only a supposition and must be improved. I suggest the addition of images obtained by AFM (preferably) or high-resolution SEM (without bacteria) presenting the real topography of Ti samples.
I understand that PI staining can be used to distinguish dead cells from alive. However, as the author stated that the coating with CICN slowed bacterial growth without killing them, this is better comprehended when a metabolic screening is performed. In this case, I suggest the inclusion of a metabolic activity experiment using MTT or other tetrazolium dyes, possibly covering washed samples without biofilm removal (to check activity without the interference of planktonic cells), the removed supernatant (planktonic cells) and without washing.
Therefore, I suggest not accepting the manuscript in its present form, and recommending a major review.
Author Response
Reviewer 1
As the authors suggest that the topography of the surface was responsible for the reduction of biofilm formation, it is of outmost importance to present a clear image of the coated surface compared to the uncoated one. Comparison with amorphous carbon coating is only a supposition and must be improved. I suggest the addition of images obtained by AFM (preferably) or high-resolution SEM (without bacteria) presenting the real topography of Ti samples.
Thank you for this feedback. We have previously considered the use of AFM to generate topographical data for the CICNT surface, but determined that it would be too difficult to get accurate measurements with the AFM equipment available to us due to both the tendency of AFM to compress the nanotubes, introducing a great deal of error, as well as the roughness of the substrate itself, which would add to the error. Instead, we have gathered new SEM images and created a new figure (Fig 1) that has the previous, top-down SEM images, as well as new images gathered of the Ti, Carbon, and CICNT surfaces from an 80 degree tilt. These images clearly show topographical differences, demonstrating that the CICNT surface has greater surface roughness while the Ti and Carbon surfaces are nearly indistinguishable from one another.
I understand that PI staining can be used to distinguish dead cells from alive. However, as the author stated that the coating with CICN slowed bacterial growth without killing them, this is better comprehended when a metabolic screening is performed. In this case, I suggest the inclusion of a metabolic activity experiment using MTT or other tetrazolium dyes, possibly covering washed samples without biofilm removal (to check activity without the interference of planktonic cells), the removed supernatant (planktonic cells) and without washing.
We sincerely appreciate this thoughtful comment and suggestion. We previously attempted metabolic screening on the sample surfaces but encountered significant variability, indicating that our protocols required further optimization. While we recognize the value of these experiments, we believe that refining the methodology would require substantial additional time, making it impractical for inclusion in this study due to the rapid turnaround time requested by the journal. However, we plan to pursue these analyses in future work. To acknowledge this, we have added the following statement to the end of the discussion at line 316:
“Although much research in this field remains to be done, it is plausible that surface texture could induce cellular changes that reduce S. aureus biofilm growth. There are some limitations in the current work. Notably, CFU quantification only accounts for metabolically active cells, potentially overlooking dormant or non-culturable populations. Additionally, as these experiments were performed in vitro, the CICNT surface may behave differently in vivo. Future studies will address these issues and further investigate the mechanisms of the CICNT antibiofilm effect. Metabolic activity studies could be used to confirm a reduction in cell growth. Transcriptomic analyses of cells grown on CICNT surfaces and bare Ti surfaces could reveal transcription-level changes to cell growth and activity. Flow cytometry could be used to confirm differences in cell division rates, while microscopy studies could also help to better understand whether specific regions of the CICNT surface preferentially support biofilm growth. This is an open area of research that merits further investigation.”
Reviewer 2 Report
Comments and Suggestions for Authors
Dear authors and editors,
the manuscript, submitted for revision, explores a highly relevant topic that deserves attention related to orthopedic implants significantly improving the quality of life of patients after prosthetic interventions. Something more implantable device-associated infections, mainly caused by biofilms, pose a serious threat and are difficult to treat due to their resistance to the immune response and antibiotics. Nanotextured surfaces are particularly suitable for reducing biofilm formation, and in particular carbon-infiltrated carbon nanotubes (CICNTs).
Nevertheless, I would like to point out that the manuscript has some weaknesses that need to be clarified, rewritten, or corrected.
Critical points to the authors.
- Line 28-32. I recommend using more up-to-date scientific literature on the subject.
- Line 89. Please specify the incubation time.
- Line 90. Washing is usually performed at least three times to ensure the removal of non-adherent bacteria.
- Line 95. In how many replicates per sample (agar plates) was the study conducted? Please describe.
- Line 95-96. In the initial cultivation, you used TSB medium, which is recommended for S. aureus growth. However, for CFU determination, you changed the medium. On what basis? It would be advisable to conduct the experiment on TSA.
- Section 3.2: Note that the SEM images are related to the observation of the structure of the investigated surfaces before biofilm formation. In this regard, and in connection with this section, it would be valuable to prepare SEM samples with cultivated biofilms on some of the test surfaces to confirm the obtained data. This analysis would also provide further clarity on the adhesion capability of the tested surfaces.
- Line 176-179. Avoid repetitions, as this has already been mentioned in Section 3.1. You can rephrase it. Additionally, the results from Figure 2 (B) for the JE2 strain appear identical to those in Figure 1, despite the assumption that this should be a separate experiment, considering that new surfaces—Al2O3-200 coated Ti or Al2O3-coated CICNT—are being analyzed.
- In relation to the analyses in Section 3.2, I would like to point out the lack of description of the materials and methodology for sample preparation for scanning electron microscopy.
- Line 240. Biofilm growth is usually measured as a quantitative value through spectrophotometric analysis. Additionally, the applied CFU test is not the most reliable, as it only accounts for metabolically active planktonic cells and the biofilm formed after cultivation on the tested surfaces. Therefore, it would be more accurate to avoid using this term. Furthermore, biofilms contain metabolically inactive cells, persister cells, etc. In such cases, the gold standard of the crystal violet assay is recommended.
- Line 241. One hour is too short a time interval to make such conclusions.
Author Response
Reviewer 2
- Line 28-32. I recommend using more up-to-date scientific literature on the subject.
Thank you for this suggestion; we have added two references. One describes the projected increase in total hip arthroplasty and total knee arthroplasty to Medicare patients in 2030 (Schichman et al. 2023), and another is a more up to date reference describing the difficulty in treating biofilm infections due to increased antibiotic resistance (Sharma et al. 2023).
- Line 89. Please specify the incubation time.
We have added the following: “Briefly, after 36 hours (unless otherwise indicated) of incubation in 24-well plates,” now found on line 101.
- Line 90. Washing is usually performed at least three times to ensure the removal of non-adherent bacteria.
Thank you for this reminder. While developing our protocol we did perform experiments to confirm that one wash was sufficient. We found that there was no significant difference in the number of unattached cells quantified from samples with 1, 2, or 3 washes, but the variability in the quantification of unattached and attached cells was much smaller with one wash, so that was the protocol we used.
- Line 95. In how many replicates per sample (agar plates) was the study conducted? Please describe.
We apologize that this section was unclear. We have added the following to line 108: “Results represent the average of three independent experiments, each consisting of biofilms grown on two samples per surface type. Each disturbed biofilm was diluted and plated onto a single agar plate, resulting in a total of six replicates per condition to ensure reproducibility.”
When developing the protocol we had in the past used three agar plate replicates per sample, and found that there was little variability in the results, so we began using one plate per sample but multiple different samples for each condition. This resulted in still having many replicates for each condition to ensure reproducibility.
- Line 95-96. In the initial cultivation, you used TSB medium, which is recommended for S. aureus growth. However, for CFU determination, you changed the medium. On what basis? It would be advisable to conduct the experiment on TSA.
Thank you for pointing this out. We have attempted to explain on lines 139-143 our rationale for using an atypical medium (RPMI) for these experiments as we attempt to move closer to an environment that is more physiologically relevant to human applicants. It has been reported that S. aureus biofilm compositions vary based upon the composition of the growth medium used, so inhibition of biofilms grown in RPMI may be different than what is seen in TSB.
In terms of the use of LB plates, we used this method since it had been used successfully before, and we have added two citations describing this method (Missiakas and Schneewind, 2013) and (Ball et al. 2022).
- Section 3.2: Note that the SEM images are related to the observation of the structure of the investigated surfaces before biofilm formation. In this regard, and in connection with this section, it would be valuable to prepare SEM samples with cultivated biofilms on some of the test surfaces to confirm the obtained data. This analysis would also provide further clarity on the adhesion capability of the tested surfaces.
We appreciate this feedback. We have previously performed SEM qualitative analysis on cultivated biofilms on test surfaces and found that the SEM images confirmed the CFU analysis findings. We have added language directing the reader to our previous publication to view these images at line 168: “In our previous study using TSB as our bacterial growth media, we had found that there was a significant 2.5-fold reduction in the number of adherent bacteria from the JE2 strain on CICNT compared to Ti, which was confirmed using both CFU counts and SEM images of the biofilm. Additionally, we found that there was a small but significant 1.5-fold reduction in the number of adherent bacteria on the Carbon control compared to Ti (Bowden et al. 2023).”
- Line 176-179. Avoid repetitions, as this has already been mentioned in Section 3.1. You can rephrase it. Additionally, the results from Figure 2 (B) for the JE2 strain appear identical to those in Figure 1, despite the assumption that this should be a separate experiment, considering that new surfaces—Al2O3-200 coated Ti or Al2O3-coated CICNT—are being analyzed.
Thank you for this feedback. We have adjusted the lines to remove repetition: “This suggests that the surface topography itself, rather than the chemistry of the material present at the bacteria-surface interface, may be reducing biofilm growth.”
In addition, you are correct to point out that the results for the JE2 strain on CICNT/Carbon/Ti are the same as in Figure 1. We had included them for reference, as described in the figure legend, but have now removed them to avoid any ambiguity.
- In relation to the analyses in Section 3.2, I would like to point out the lack of description of the materials and methodology for sample preparation for scanning electron microscopy.
We appreciate this correction. We have added a section to line 126 to the methods for electron microscopy: “Scanning electron microscopy (SEM) images were taken on a ThermoScientific Verios G4 UC SEM. Samples of each surface type were imaged prior to biofilm formation using either a flat stub at an accelerating voltage of 5 kV and ion beam current of 50 pA or at an 80 degree tilt with an accelerating voltage of 10 kV and an ion beam current of 0.1 nA.”
- Line 240. Biofilm growth is usually measured as a quantitative value through spectrophotometric analysis. Additionally, the applied CFU test is not the most reliable, as it only accounts for metabolically active planktonic cells and the biofilm formed after cultivation on the tested surfaces. Therefore, it would be more accurate to avoid using this term. Furthermore, biofilms contain metabolically inactive cells, persister cells, etc. In such cases, the gold standard of the crystal violet assay is recommended.
Thank you for pointing this out. When we began working with the CICNT surface we at first did use crystal violet. However, we quickly discovered that crystal violet is not the best choice for CICNT surfaces because the dye gets caught in the nanotubes, and even with stain controls the amount of background staining is often too high to be able to articulate conclusions. This makes CFU counting a dependable method even though it does possess limitations such as only counting metabolically active cells. We added the following to the methods section on biofilm quantification, starting on line 96:
“Multiple methods of biofilm quantification were considered, including crystal violet and CFU quantification. Crystal violet is often considered the gold standard method, but the dye becomes trapped in the porous nanotubes on the CICNT surface, producing substantial background staining that confounds the results. Therefore, we used serial dilutions and colony forming unit (CFU) quantification.”
- Line 241. One hour is too short a time interval to make such conclusions.
We apologize for being unclear. We agree that one hour is too short, and so these conclusions are based on data collected at later time points (especially 36 hours). We have clarified the point at line 289: “...there is a decrease in planktonic cells as well as adherent cells on the CICNT samples at 36 hours. One possible explanation for this could be that certain areas of the CICNT surface are more amenable to bacterial growth than others...”
Reviewer 3 Report
Comments and Suggestions for Authors
The paper entitled “Carbon-Infiltrated Carbon Nanotube Topography Reduces the Growth of Staphylococcus aureus Biofilms” by Lucy C. Bowden, Sidney T. Sithole, Anton E. Bowden, Brian D. Jensen and Bradford K. Berges describe the role of the topography of the CICNT and its surface chemistry in the reduction in biofilm growth. I believe that the results are interesting for Nanomaterials and the manuscript can be accepted for publications after minor revision.
- Please, highlight the goal of the present study.
- Please, highlight the novelty of the results.
- In conclusion, please, describe the limitations and remaining problems and ways to solve them in your opinion.
Author Response
Reviewer 3
- Please, highlight the goal of the present study.
Thank you for these suggestions. To address both the goal of the study and the novelty of the results we have added the following to the introduction at line 46: “The goal of this study was to investigate how the CICNT surface reduces biofilm formation in a more physiologically relevant medium. Using multiple controls, we determined that the surface inhibits bacterial adhesion primarily through its topography rather than its chemical properties. Additionally, our findings suggest that the surface reduces biofilm formation by limiting bacterial growth rather than by killing the cells or preventing initial attachment. This study introduces a novel surface with anti-biofilm properties that may uniquely inhibit S. aureus biofilm formation.”
- Please, highlight the novelty of the results.
We have also clarified in the discussion the novelty of the results beginning at line 263: “Commonly, materials exhibiting structural antibiofilm properties are considered in one of two ways. They may be bactericidal (killing the bacteria on contact), or antibiofouling, which is generally defined as the prevention of bacterial attachment (Hasan et al. 2013). For example, many insect wings exhibit bactericidal properties (Ivanova et al. 2012; Bandara et al. 2017), while shark skin is considered to be antibiofouling (Rostami et al. 2022). However, the CICNT surface does not fit neatly into these two categories, and may act in a more novel manner. Flow cytometry data indicates that the bacteria are not dying more rapidly on a CICNT surface than on a Ti surface, and CFU data indicates that at 1 hour and 12 hours there is no difference in the number of bacteria on the CICNT surface and a Ti surface, suggesting no difference in initial attachment rates. Rather than being bactericidal or antibiofouling, the CICNT surface may fall into a less-explored third category, where the surface topography restricts bacterial growth without killing the bacteria even though the bacteria can effectively attach. Surfaces of this type could be referred to as bacteriostatic if they restrict bacterial growth or replication but do not kill the bacteria.
Few examples of bacteriostatic surfaces exist in the literature, but one example of a surface that reduces bacterial growth in this manner was described by Sorzabal-Bellido et al....”
- In conclusion, please, describe the limitations and remaining problems and ways to solve them in your opinion.
We appreciate this feedback. We have added the following to the conclusion beginning at line 317: “There are some limitations in the current work. Notably, CFU quantification only accounts for metabolically active cells, potentially overlooking dormant or non-culturable populations. Additionally, as these experiments were performed in vitro, the CICNT surface may behave differently in vivo. Future studies will address these issues and further investigate the mechanisms of the CICNT antibiofilm effect. Transcriptomic analyses of cells grown on CICNT surfaces and bare Ti surfaces could reveal transcription-level changes to cell growth and activity. Flow cytometry could be used to confirm differences in cell division rates, while microscopy studies could also help to better understand whether specific regions of the CICNT surface preferentially support biofilm growth. This is an open area of research that merits further investigation.”